# Characteristics and Drivers of Vegetation Change in Xinjiang, 2000–2020

Guo Li [1,2], Jiye Liang [3], Shijie Wang [4,5], Mengxue Zhou [1,2], Yi Sun [4], Jiajia Wang [4] and Jinglong Fan [4,5,*]

[1]   College of Life Science and Technology, Tarim University, Alar 843300, China; liguo163666@163.com (G.L.); sky17179@163.com (M.Z.)
[2]   Xinjiang Production and Construction Corps Key Laboratory of Protection and Utilization of Biological Resources in Tarim Basin, Alar 843300, China
[3]   School of Pharmacy, Youjiang Medical College of Nationalities, Baise 533000, China; ljy_123123@126.com
[4]   State Key Laboratory of Desert and Oasis Ecology, Xinjiang Institute of Ecology and Geography, Chinese Academy of Sciences, Urumqi 830011, China; wangshijie163@163.com (S.W.); sunyi2@ms.xjb.ac.cn (Y.S.); wangjiajia@ms.xjb.ac.cn (J.W.)
[5]   Taklamakan Desert Research Station, Xinjiang Institute of Ecology and Geography, Chinese Academy of Sciences, Koral 841000, China
[*]   Correspondence: fanjl@ms.xjb.ac.cn

**Abstract:** Examining the features of vegetation change and analyzing its driving forces across an extensive time series in Xinjiang are pivotal for the ecological environment. This research can offer a crucial point of reference for regional ecological conservation endeavors. We calculated the fractional vegetation cover (FVC) using MOD13Q1 data accessed through the Google Earth Engine (GEE) platform. To discern the characteristics of vegetation changes and forecast future trends, we employed time series analysis, coefficient of variation, and the Hurst exponent. The correlation between climate factors and FVC was investigated through correlation analysis. Simultaneously, to determine the relative impact of meteorological change and anthropogenic actions on FVC, we utilized multiple regression residual analysis. Furthermore, adhering to China's ecological functional zone classification, Xinjiang was segmented into five ecological zones: R1 Altai Mountains-Junggar West Mountain Forest and Grassland Ecoregion, R2 Junggar Basin Desert Ecoregion, R3 Tianshan Mountains Mountain Forest and Grassland Ecoregion, R4 Tarim Basin-Eastern Frontier Desert Ecoregion, and R5 Pamir-Kunlun Mountains-Altan Mountains Alpine Desert and Grassland Ecoregion. A comparative analysis of these five regions was subsequently conducted. The results showed the following: (1) During the first two decades of the 21st century, the overall FVC in Xinjiang primarily exhibited a trend of growth, exhibiting a rate of increase of $4 \times 10^{-4}$ y$^{-1}$. The multi-year average FVC was 0.223. The mean value of the multi-year FVC was 0.223, and the mean values of different ecological zones showed the following order: R1 > R3 > R2 > R5 > R4. (2) The predominant spatial pattern of FVC across Xinjiang's landscape is characterized by higher coverage in the northwest and lower in the southeast. In this region, 66.63% of the terrain exhibits deteriorating vegetation, while 11% of the region exhibits a notable rise in plant growth. Future changes in FVC will be dominated by a decreasing trend. Regarding the coefficient of variation outcomes, a minor variation, representing 42.12% of the total, is noticeable; the mean coefficient of variation stands at 0.2786. The stability across varied ecological zones follows the order: R1 > R3 > R2 > R4 > R5. (3) Factors that have a facilitating effect on vegetation FVC included relative humidity, daylight hours, and precipitation, with relative humidity having a greater influence, while factors that have a hindering effect on vegetation FVC included air temperature and wind speed, with wind speed having a greater influence. (4) Vegetation alterations are primarily influenced by climate change, while human activities play a secondary role, contributing 56.93% and 43.07%, respectively. This research underscores the necessity for continued surveillance of vegetation dynamics and the enhancement of policies focused on habitat renewal and the safeguarding of vegetation in Xinjiang.

**Keywords:** vegetation FVC; residual analysis; climate change; human activities

## 1. Introduction

In land-based ecosystems, vegetation plays a crucial role in the interchange of water and energy between the soil surface and the atmosphere, and it is also vital for carbon cycling. This contributes substantially to lessening the influence of climate change on natural ecosystems [1,2]. With the escalation of global warming, addressing the hazards associated with climate change and improving ecosystem management will gain heightened importance [3,4]. Enhancing the monitoring of vegetation dynamics, a key indicator of both global climate change and responses in ecosystem management, is crucial. It aids in a more comprehensive analysis of ecosystem structure and function, thereby fostering ecosystem sustainability [5–7]. Within the realm of Vegetation Index, the normalized difference vegetation index (NDVI) is notably the most prevalent. However, its practical use is often hindered by several factors, including atmospheric noise, soil background interference, saturation in the red-light spectrum, and various other issues [8]. Fractional Vegetation Cover (FVC) indicates the fraction of the ground area vertically projected by vegetation [9]. It also serves as a crucial metric for tracking alterations in the ecological environment [10]. Additionally, when observing dynamic shifts in vegetation cover within regions affected by wind-blown sand and in sandy areas, FVC demonstrates superior quantification capabilities [11]. Extensive research indicates that the primary factors affecting vegetation changes are frequently linked to meteorological change and anthropogenic actions [12]. Research has concentrated on scientifically identifying and measuring the impact of climate change and anthropogenic factors on the coverage of vegetation [13]. Measuring the impact of both climatic shifts and anthropogenic endeavors on vegetation dynamics is vital for comprehending the interplay among ecosystems, the atmosphere, and human communities. This understanding is pivotal for formulating effective ecological rehabilitation approaches.

The alteration in vegetation is impacted by an interplay of climatic variations and human interventions. Global warming's elevated temperatures and altered precipitation patterns can markedly influence changes in vegetation distribution and behavior [14,15]. Precipitation and temperature are frequently emphasized as key climatic elements focused on in research that influence vegetation dynamics, and these factors significantly impact the growth of vegetation, as well as its photosynthesis and respiration processes [16–19]. In addition, other scholars have suggested that elements like daylight hours and wind velocity also have notable bearing on the increase of plant growth [20,21]. However, in a significant proportion of current studies, there has been a preference for analyzing only the correlations between vegetation indices and meteorological factors to ascertain the effect of climatic factors on alterations in vegetation [22,23]. Human-induced factors, covering the introduction of green initiatives [24], land use planning [25], grazing activities [26], and the advancement of urbanization [27], also affect vegetation changes to varying degrees. From the early 2000s, China's government has focused on ecological capital protection and restoration [28], especially for areas such as northern China [29], the Yangtze River Basin [30], and the intermediate and higher areas of the Yellow River Basin [31]. As a result, implementation programs for ecosystem management have also had a broad impact on vegetation change in China [25]. Vegetation dynamics are driven by a variety of intricate factors, meteorological shifts, and anthropogenic interventions, particularly in ecosystem management, which stand out as significant contributors to these changes [32]. The task of quantifying vegetation's reaction to human influences and climate variation remains an area requiring additional exploration.

Understanding how climate fluctuations and anthropogenic actions influence vegetation dynamics contributes to a more scientific examination of the factors driving these alterations and assists in offering strategies for effective and sustainable environmental management [16,33]. Numerous earlier studies have primarily utilized linear regression and correlation analysis techniques to examine the impacts of climate elements and human activities on the dynamics of vegetation; yet, linear analyses may be subject to errors [34,35]. Among the various techniques for attributing changes in vegetation, residual trend analysis is widely regarded as efficient. It quantifies the impact of both climate change and human

actions on vegetation alterations. This is achieved by establishing a linear association between the remote sensing index of vegetation and climatic elements, subsequently removing the climatic impact on vegetation alterations, and thereafter assessing the human contributions to these changes [36]. However, current studies on residual analysis have focused excessively on the influence of precipitation and temperature [37,38].

Xinjiang, located in the arid zone of northwestern China, is undergoing significant impacts due to climate change; since the twenty-first century, as the "One Belt, One Road" initiative in Xinjiang has continued to advance, anthropogenic actions have had an increasing impact on the vegetation cover, and activities such as over-cultivation, over-grazing, and indiscriminate logging have exacerbated the environmental problems. Despite the Chinese government's implementation of active ecological policies such as protecting natural forests, initiatives to convert farmland back to forests and grasslands, and rotational grazing, Xinjiang's ecological problems remain significant. At the same time, as a result of global warming, Xinjiang's climate is changing to a "warmer and wetter" climate, with a marked increase in precipitation, but the positive benefits of increased precipitation are offset by higher temperatures, further complicating Xinjiang's climate problems. Therefore, in this research, we estimated the FVC based on different ecological functional zones, and we used NDVI data to represent the dynamic changes of vegetation. Our study aimed to (1) investigate the changes in spatial and temporal dimensions of Xinjiang's FVC from 2000 to 2020; (2) determine the impact of weather conditions on the variations in FVC; (3) and assess the influence of climatic alterations as well as human interventions on the variations in Xinjiang's vegetation FVC. Therefore, monitoring Xinjiang's ecological environment and thoroughly considering the consequences of both environmental change and anthropogenic endeavors on its vegetation FVC is fundamental for promoting the region's ecological well-being and ensuring its ecological security.

## 2. Materials and Methods

### 2.1. Overview of the Study Area

Positioned between 73°40′ and 96°23′ E, and 34°22′ and 49°10′ N, Xinjiang lies at China's northwestern frontier, marking the intersection of Central and East Asia. Xinjiang has a complex topography and geomorphology, with large elevation differences within its borders (Figure 1b), a pronounced temperate continental climate, large temperature differences, sufficient sunshine hours, and sparse rainfall of about 150 mm per year. Vegetation types in Xinjiang are very complex with large regional differences. With the implementation of national key restoration projects [39], the land use and cover pattern in Xinjiang has been greatly changed. China's varied bioclimatic regions, combined with intense ecosystem management disturbances, present an excellent chance to define the impacts of meteorological change and sustainable ecosystem governance on vegetation behavior. Owing to the substantial spatial variability in vegetation distribution governed by climatic elements and ecosystem management, vegetation growth responses could vary across different areas [40]. Ecogeographic zoning provides a method to explore the differentiation in vegetation response [41].

### 2.2. Data Sources

As we can see from Table 1, MODIS vegetation information was derived from NASA's MOD13Q1 dataset, accessible via the MODIS Land Processes Distributed Active Archive Center (https://ladsweb.modaps.eosdis.nasa.gov/, accessed on 18 June 2023). This dataset features a spatial resolution of 250 m and a 16-day temporal resolution, covering between February 2000 and December 2020. The data underwent preprocessing, which included format transformation, reprojection, and merging using the MRT (MODIS Reprojection Tools) software. Additionally, we applied Savitzky-Golay filtering in Matlab 2022b to remove mixed noise from the images and enhance the NDVI bands' quality [42,43]. Using the Google Earth Engine (GEE) platform [44], atmospheric, cloud, and solar zenith angle effects were mitigated using the median composite method, resulting in monthly NDVI

data. For the vegetation growth season (April–September) that was selected, the mean synthesis was carried out to obtain the annual-scale MODIS-FVC dataset of Xinjiang using ArcGIS 10.8 software with the image-element dichotomization method.

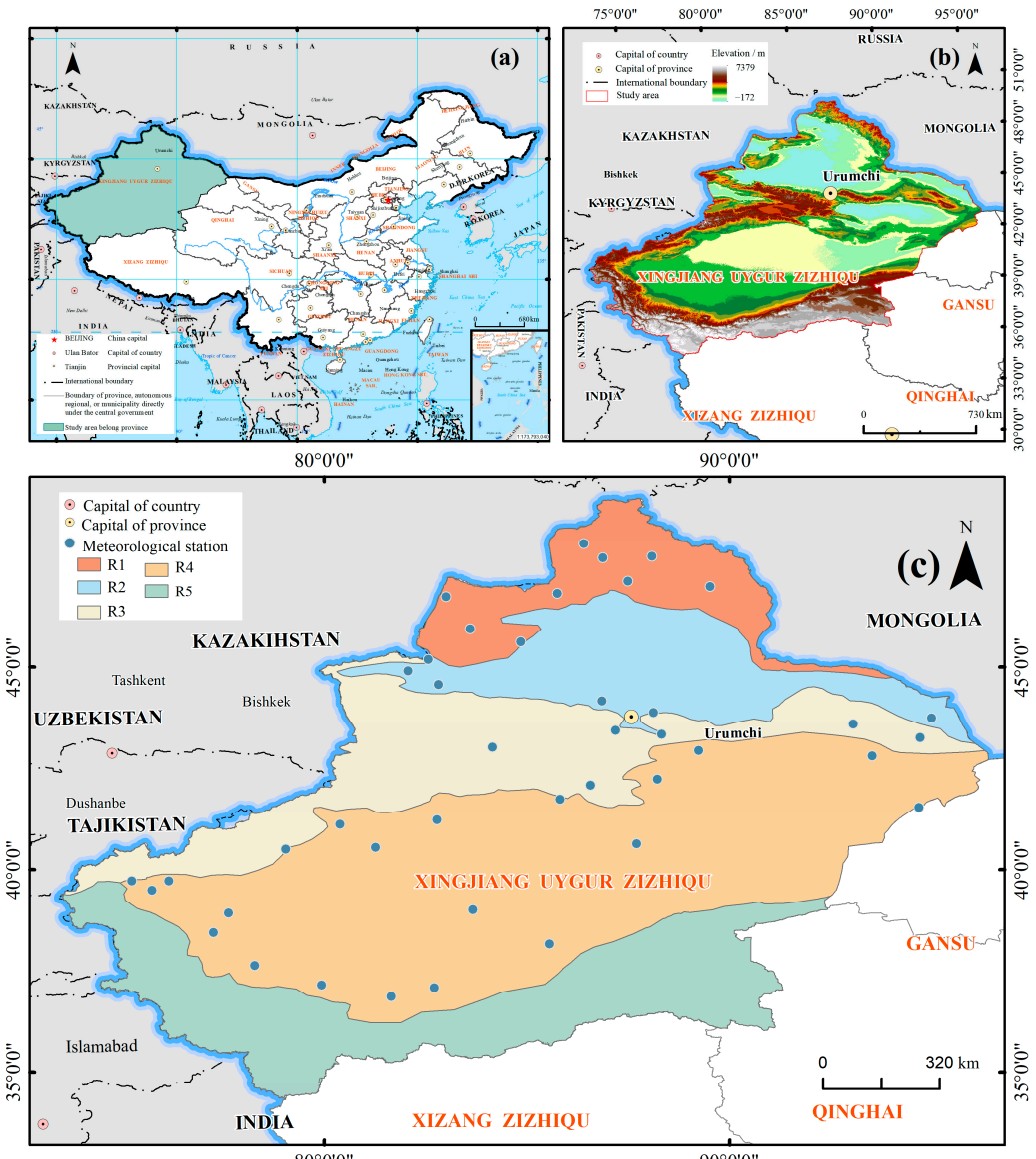

**Figure 1.** (**a**) Geographic position of the research area. (**b**) Topographic map of the study area. (**c**) Distribution of ecological functional areas. Note: the map used here adheres to the standard map, acquired under review number GS (2019) 1822 from the Ministry of Natural Resources Standard Map Service website, and retains the original map boundaries unaltered. R1: Altai Mountains-Junggar West Mountain Forest and Grassland Ecoregion, R2: Junggar Basin Desert Ecoregion, R3: Tianshan Mountains Mountain Forest and Grassland Ecoregion, R4: Tarim Basin-Eastern Frontier Desert Ecoregion, and R5: Pamir-Kunlun Mountains-Altan Mountains Alpine Desert and Grassland Ecoregion.

In this study, we sourced meteorological information from the China Meteorological Data Service Center's daily values dataset (https://data.cma.cn/, accessed on 5 March 2023), and 42 meteorological stations within the territory were selected for the period of 2000–2020, including the five elements of wind speed, sunshine, relative humidity, temperature, and precipitation. We converted the daily records into monthly aggregates. Using the thin plate smoothing spline function in Anusplin 4.2 along with spatial interpolation techniques, we generated raster data. This data matched the vegetation cover FVC in both temporal resolution and projection [45].

| Data Set | Timing | Time Resolution | Spatial Resolution | Source |
|---|---|---|---|---|
| MODIS | 2000−2020 | 16 days | 250 m | https://ladsweb.modaps.eosdis.nasa.gov/ (accessed on 18 June 2023) |
| Meteorological data | 2000−2020 | 30 days | 250 m | http://data.cma.cn (accessed on 5 March 2023) |
| DEM | / | / | 250 m | http://www.gscloud.cn (accessed on 11 December 2023) |

*2.3. Methodology*

2.3.1. Trend Analysis of FVC Changes Based on Image Elements

We utilized the Theil–Sen median trend method and analysis for identifying the trend in the FVC time series and to measure the rate at which the FVC changed [46,47], which was calculated as follows:

$$\beta = median\left(\frac{x_i - x_j}{i - j}\right), \forall j > i \tag{1}$$

Among them, $x_i$ and $x_j$ represent the sequential data, with $i$ and $j$ being representative of time series data. While $\beta > 0$ indicates an upward trajectory in the time series, a value less than 0 suggests a downward trajectory.

This paper employed the Mann–Kendall technique to ascertain the statistical significance of the observed trend [48], calculated as follows:

$$S = \sum_{i}^{n-1} \sum_{j=i+1}^{n} sign\left(x_j - x_i\right) \tag{2}$$

$$sign\left(x_j - x_i\right) = \begin{cases} 1, & x_j - x_i > 0 \\ 0, & x_j - x_i = 0 \\ 1, & x_j - x_i < 0 \end{cases} \tag{3}$$

$$Z_c = \begin{cases} \frac{S-1}{\sqrt{Var(S)}}, & if\ S > 0 \\ 0, & if\ S = 0 \\ \frac{S+1}{\sqrt{Var(S)}}, & if\ S < 0 \end{cases} \tag{4}$$

In the formula $S$ represents the test statistic; $Z_c$ is the normalized test statistic; $x_i$ and $x_j$ are the time series data; and $n$ refers to the total count of samples in the series.

$$Var(S) = \frac{n(n-1)(2n+5)}{18}$$

In conjunction with the trend indicated by the $\beta$ value, the importance of the FVC trend alteration was evaluated against significance indicators. Significance levels $P_{0.01}$ and $P_{0.05}$ were chosen as the pivotal values, leading to their categorization into six distinct groups, as illustrated in Table 2 [49].

**Table 2.** Trend analysis rating scale.

| Trend of the FVC | $\beta$ | $Z_c$ | Trend of the FVC | $\beta$ | $Z_c$ |
|---|---|---|---|---|---|
| Highly significant reduction | $\beta < 0$ | $|Z_c| > 2.58$ | No significant increase | $\beta \geq 0$ | $1.96 > |Z_c| > 0$ |
| Significant reduction | $\beta < 0$ | $2.58 \geq |Z_c| \geq 1.96$ | Significant increase | $\beta \geq 0$ | $2.58 \geq |Z_c| \geq 1.96$ |
| No significant reduction | $\beta < 0$ | $1.96 > |Z_c| > 0$ | Highly significant increase | $\beta \geq 0$ | $|Z_c| > 2.58$ |

### 2.3.2. Coefficient of Variation of the Vegetation FVC Based on Image Elements

The fluctuation of vegetation cover in the time series was responded to by calculating the coefficient of variation $C_v$ of the FVC from 2000 to 2020 in Xinjiang [50]. A large value of the coefficient of variation indicates that the vegetation is subjected to a greater intensity of disturbance, which means that it is more unstable.

$$C_v = \sqrt{\frac{1}{n-1}\sum_{i=1}^{n}(FVC_i - FVC)^2} / FVC$$

### 2.3.3. Image Element-Based FVC Persistence Analysis

Previous research has demonstrated that the Hurst index is capable of quantitatively depicting the reliance of variables in time series and assessing their temporal trends [51]. The special computations method can be found in related studies, and the formula is as follows:

$$\frac{R(T)}{S(T)} = (mT)^H$$

$$R(T) = \max_{1 \le t \le T} X(t, T) - \min_{1 \le t \le T} X(t, T)$$

$$S(T) = \sqrt{\frac{1}{T}\sum_{t=1}^{T}\left(FVC_T - \overline{FVC_T}\right)^2}$$

$$X(t, T) = \sum_{t=1}^{T}\left(FVC_t - \overline{FVC_T}\right)$$

$$\overline{FVC_T} = \frac{1}{T}\sum_{t=1}^{T} FVC_x$$

In this context, the Hurst exponent is denoted by $H$, with $R(T)$ representing the series of extreme deviations. The series of standard deviations is symbolized by $S(T)$, whereas $m$ represents a constant value of 1. The cumulative deviation is expressed as $X(t, T)$, and $FVC_t (t = 1, 2, \ldots, n)$ signifies the time series of $FVC$. Additionally, $FVC_T$ ($T = t, t + 1, \ldots, n$) is indicative of the average series.

When $H = 0.5$, it signifies that upcoming changes are independent of previous ones; when $0.5 < H < 1$, it suggests that future changes are likely to continue in the same direction as historical trends, with the proximity of $H$ to 1 indicating a higher degree of persistence. When $0 < H < 0.5$, it implies that the forthcoming trend will be the reverse of the past trend, and the closer H is to 0, the more pronounced the tendency for reversal.

### 2.3.4. Pearson Correlation Analysis Based on Image Elements

This paper utilizes correlation analysis to determine the Pearson correlation coefficient, reflecting the relationship between FVC and various meteorological elements [36]. The calculation process is as follows:

$$R_{xy} = \frac{\sum_{i=1}^{n}(x_i - \bar{x})(y_i - \bar{y})}{\sqrt{\sum_{i=1}^{n}(x_i - \bar{x})^2 \sum_{i=1}^{n}(y_i - \bar{y})^2}}$$

where $x$ and $y$ are the mean values of the two variables over $n$ years. Here, $R_{xy}$ is identified as the basic correlation coefficient linking the two factors, and $n$ stands for the size of the sample. Correlation coefficients were tested for significance using the *t*-test.

### 2.3.5. Multivariate Regression Residual Analysis

In this research, multiple regression residual analysis was employed to investigate the effects and respective contributions of human activities and climate change on the vegetation following alterations [16,52,53]. The methodology was divided into three steps:

(1) A multiple linear regression model, incorporating data on vegetation cover, air temperature, precipitation, relative humidity, daylight hours and wind speed, was constructed to determine the parameters. (2) These parameters were applied to forecast alterations in the vegetation with the $FVC_{CC}$ representing the influence of climatic factors. (3) The residual values of $FVC_{OBS}$ and $FVC_{CC}$ were calculated, with $FVC_{HA}$ signifying the impact of human activities on the vegetation. The following formula was used for this calculation:

$$FVC_{CC} = a \times Pre + b \times TEM + c \times SSD + d \times Rhu + e \times Wins + f$$

$$FVC_{HA} = FVC_{OBS} - FVC_{CC}$$

$$FVC_{slope} = \frac{n \times \sum_{i=1}^{n} (i \times FVC_i) - (\sum_{i=1}^{n} i)(\sum_{i=1}^{n} FVC_i)}{n \times \sum_{i=1}^{n} i^2 - (\sum_{i=1}^{n} i)^2}$$

where $a$, $b$, $c$, $d$ and $e$ are coefficients in the regression corresponding to temperature, precipitation, daylight hours, relative humidity, and wind speed and $f$ is a constant term. $Pre$ represents yearly precipitation, $TEM$ stands for average yearly temperature, $SSD$ is the daylight hours, $Rhu$ is the mean annual humidity, $Wins$ is the mean annual wind speed, $FVC_{CC}$ is the forecasted outcome from the multiple regression equation, $FVC_{OBS}$ is the observed actual value, $FVC_{HA}$ is the residual value, $FVC_{slope}$ is the gradient of the single variable linear regression equation, $i$ is the temporal factor, $n$ represents the duration in years, and $FVC_i$ reflects the impact value attributed to either human activities or climate change.

To better understand how both human activities and natural elements affect vegetation cover changes in the research area, Table 3 was utilized to identify the driving forces and compute their contribution percentages.

**Table 3.** Criteria for determining the drivers of changes in the vegetation FVC and methods for calculating contribution rates.

| Slope ($FVC_{OBS}$) [a] | Driving Factor | Classification Criteria | | Contribution Rate (%) | |
|---|---|---|---|---|---|
| | | Slope ($FVC_{CC}$) [b] | Slope ($FVC_{HA}$) [c] | Climate Change (CC) | Human Activity (HA) |
| >0 | CC&HA | >0 | >0 | $\frac{slope(FVC_{CC})}{slope(FVC_{obs})}$ | $\frac{slope(FVC_{CC})}{slope(FVC_{obs})}$ |
| | CC | >0 | <0 | 100 | 0 |
| | HA | <0 | >0 | 0 | 100 |
| <0 | CC&HA | <0 | <0 | $\frac{slope(FVC_{CC})}{slope(FVC_{obs})}$ | $\frac{slope(FVC_{CC})}{slope(FVC_{obs})}$ |
| | CC | <0 | >0 | 100 | 0 |
| | HA | >0 | <0 | 0 | 100 |

The symbols 'a', 'b', and 'c' are used without any particular meaning or numerical significance. They are employed to distinguish between actual values, predicted values, and the values of residuals.

## 3. Results

### 3.1. Characteristics of Spatial and Temporal Dynamics of the FVC in Xinjiang

The trend of interannual growing season vegetation FVC changes in Xinjiang and different ecological functional areas are shown in Figure 2. Between 2000 and 2020, there was an observable increase in the average FVC throughout Xinjiang's cultivation period, escalating from 0.216 to 0.222, showing a significant fluctuating growth trend in general, with a multi-year growth rate of 2.79%, a growth rate of 0.004/10a, and an average value of the FVC of 0.223 over multiple years. The peak FVC was recorded in 2017, while the lowest was in 2000.

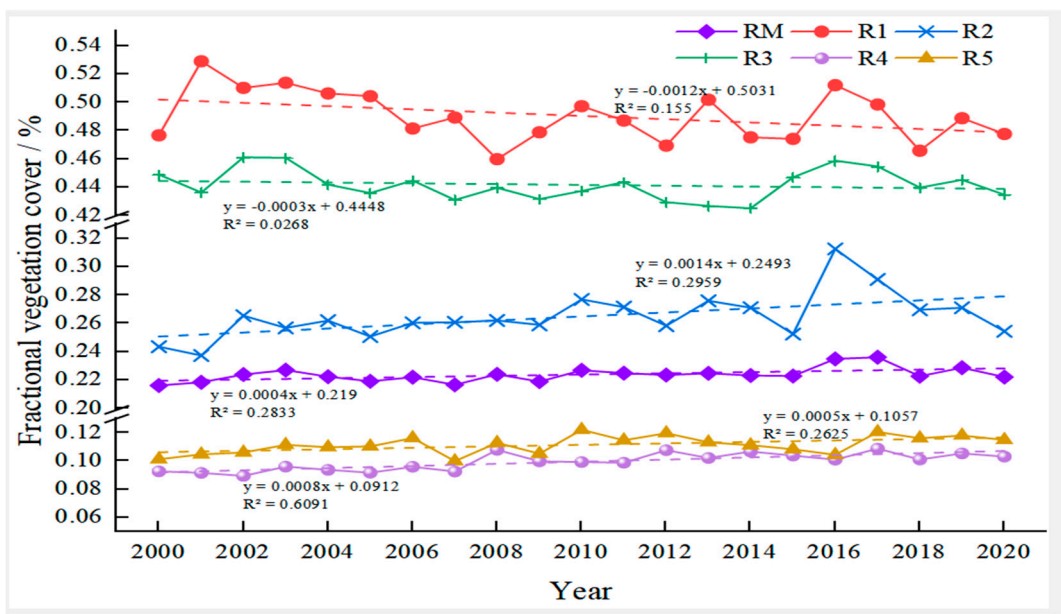

**Figure 2.** Spatial and temporal variation characteristics of the FVC in Xinjiang, 2000–2020.

There were regional differences in FVC changes among different ecological function zones in Xinjiang, with R2 (0.014/10a) > R4 (0.008/10a) > R5 (0.005/10a) > R3 (−0.003/10a) > R1 (−0.012/10a). R1 and R3 had better original vegetation conditions, with multi-year average values of 0.490 and 0.442, respectively. These figures stood significantly above the general level and those of other varied ecological function areas in Xinjiang, but the vegetation FVC showed a declining trend. While R1 and R3 initially had favorable conditions, as evidenced by their growing season FVC averages of 0.490 and 0.442, these were much higher than that of Xinjiang as a whole and other ecological function zones. However, the vegetation FVC showed a declining trend, and the development speed of the vegetation FVC in R2, R4, and R5 was higher than that of Xinjiang as a whole (0.004/10a).

Figure 3a illustrates that the overall spatial distribution of vegetation FVC in Xinjiang predominantly follows a pattern of being higher in the northwest and lower in the southeast. The region is largely characterized by very low vegetation coverage, comprising 68.55% of its total area, while the combined areas of medium, high, and very high FVC coverage constitute just 18.42% of Xinjiang's total land. According to Figure 3b, the variation rate of FVC in Xinjiang spans from −0.059 to 0.065, demonstrating significant spatial variation in the FVC trends. Notably, areas showing a declining trend in vegetation cover represent 67.52% of Xinjiang's total area.

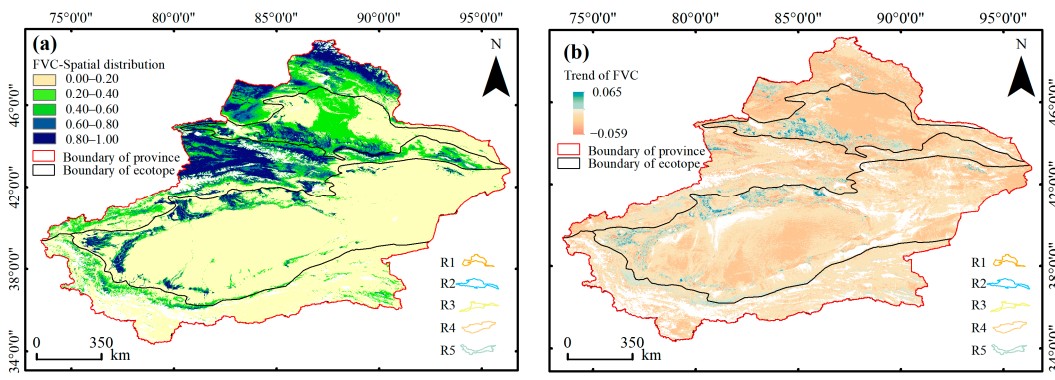

**Figure 3.** (**a**) Spatial distribution of the FVC, 2000–2020; (**b**) annual trend of the FVC, 2000–2020.

By examining the trend of the vegetation FVC across various ecological functional zones and overlaying the M-K significance values (as shown in Figure 4), the spatial

distribution trend of Xinjiang's vegetation FVC and its different ecological zones was determined. From Figure 4, it is observable that the predominant trend in the variation of vegetation FVC in Xinjiang was dominated by an insignificant decrease, accounting for 42.89%; this was an insignificant increase that accounted for 24.24%, indicating that the vegetation FVC status in Xinjiang was degraded, and the area of improvement was gradually decreasing. The areas of R4, R2, and R1 experienced pronounced degradation, which was centrally concentrated, while the region marked as R5 was primarily where a significant increase in vegetation was observed.

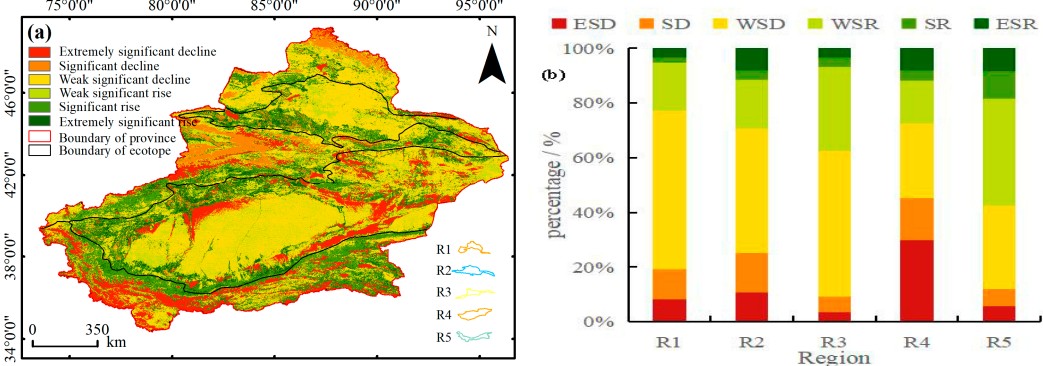

**Figure 4.** (**a**) Significance of the trend of FVC change from 2000 to 2020; (**b**) statistical map of the significance of the trend of change in different ecological functional zones. Note: ESD stands for highly significant decrease, SD stands for significant decrease, ESD stands for non-significant decrease, WSR stands for non-significant increase, SR stands for significant increase, and ESR stands for highly significant increase.

### 3.2. Spatial Distribution of Vegetation FVC Stability in Xinjiang

An analysis was conducted on the coefficient of variation (CV) for Xinjiang's FVC from 2000 to 2020, as depicted in Figure 5. Variation coefficients within 0–0.15 were categorized as weak, 0.15–0.30 as medium variation, and above 0.30 as strong variation. In Xinjiang's vegetation, the variation coefficients ranged from 0 to 4.56, with weak variation accounting for 42.12% of the total, medium variation accounting for 38.82% of the total, and strong variation accounting for 19.06% of the total. The mean variation coefficient stood at 0.2786, indicating a steady trend of stabilization moving from the Tien Shan Mountains (R3) and Altai Mountains (R1) towards the Junggar Basin (R2) and Tarim Basin (R4), with the general condition being relatively stable.

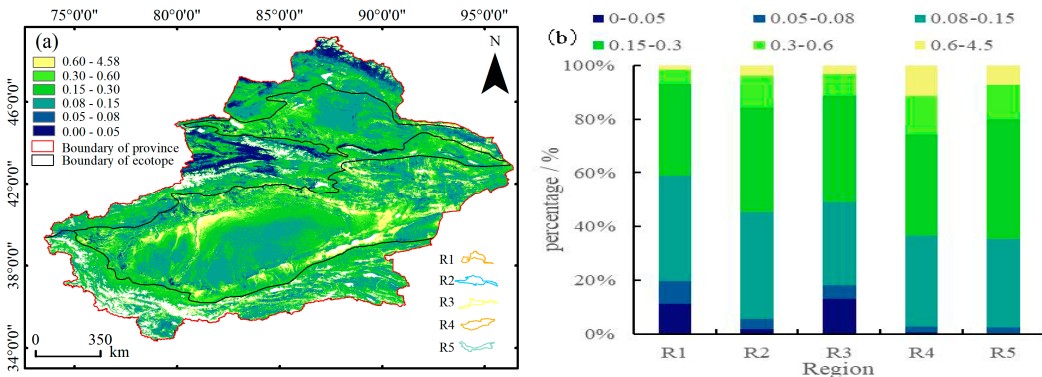

**Figure 5.** (**a**) Spatial distribution of the FVC coefficient of variation, 2000–2020; (**b**) statistical map of different ecological functional zones of the FVC coefficient of variation, 2000–2020.

### 3.3. Spatial Distribution and Future Trends of Hurst Index in Xinjiang

Calculating the Hurst index of FVC in Xinjiang, in conjunction with the Theil–Sen trend analysis, facilitated predictions about the future trajectory of FVC both in Xinjiang

and its various ecological function subregions, as shown in Figure 6a. Between 2000 and 2020, the Hurst index for vegetation cover in Xinjiang ranged from 0.027 to 1, with an average of 0.4514. Notably, 72.45% of the total area had a Hurst index exceeding 0.5, primarily located in the northern and southern regions of the Tianshan Mountains and the northern Kunlun Mountains. Conversely, 27.54% of the area had a Hurst index below 0.5. The projected future changes were inversely related to past trends, with lower values predominantly in the Gurbantunggut Desert. This suggests that the positive continuity in vegetation change in Xinjiang was more pronounced than the negative continuity.

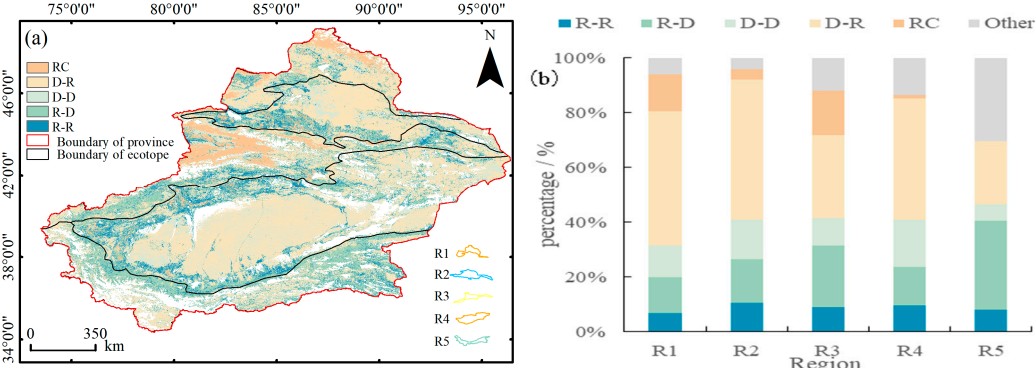

**Figure 6.** (**a**) Spatial distribution of future trends in the FVC from 2000 to 2020; (**b**) statistical map of future trends in different ecological functional zones. Note: R-R stands for continuous improvement, R-D stands for increasing to decreasing, D-D stands for continuous degradation, D-R stands for decreasing to increasing, R-C stands for random change, and Other stands for failing the test.

By integrating the Hurst index with the FVC change trend in Xinjiang during 2000–2020, the whole area can be classified into six unique categories: persistent improvement (R-R), growth followed by decline (R-D), continuous decline (D-D), decline followed by growth (D-R), random fluctuation (RC), and inconclusive results (Other), as depicted in Figure 6b. Areas showing a persistent improvement in FVC, covering 9.00% of the total area, are primarily located in oasis cities to the east of the Taklamakan and Gurbantunggut deserts. These densely populated regions can adversely affect the local land and ecosystems, leading to prolonged land instability and making them critical for ecological conservation and management. Regions exhibiting a sustained decline in FVC, indicative of ongoing vegetation reduction, comprise 13.16% of the total area. Areas where FVC initially increased but are projected to decrease, signifying a shift from growth to degradation, make up 18.83% of the land, particularly noticeable in R5. Conversely, areas transitioning from a declining to an improving FVC, hinting at ecological recovery, constitute 39.44% of the total area and are especially prevalent in R1, R2, and R4. Random variation, where future trends are unrelated to past patterns, accounts for 5.26% of the total area.

### 3.4. Correlation Analysis between the Vegetation FVC and Climate Factors in Xinjiang

Weather elements significantly influence the FVC of vegetation, particularly factors like sunshine duration, wind speed, precipitation, relative humidity, and temperature. Figure 7 illustrates the trends over time for these climatic elements from 2000 to 2020. It reveals that both rainfall and temperature have been on an upward trajectory, increasing at rates of 1.0013 mm per year and 0.029 °C per year, respectively. Conversely, trends for relative humidity, sunshine duration, and wind speed were downward, with change rates of −0.1% per year, −0.0171 h per year, and −0.0016 m per second per year, respectively.

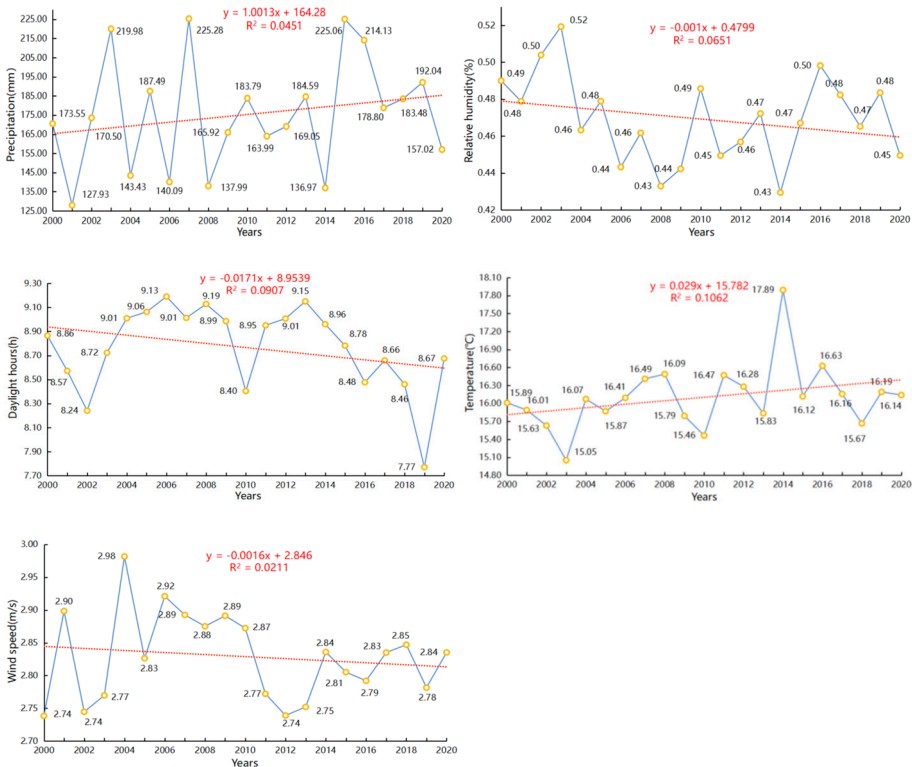

**Figure 7.** Characteristics of meteorological factor changes in Xinjiang between the beginning of 2000 and the close of 2020. Note: The red dashed line represents the linear fit, the blue solid line represents the trend line, the red font represents the linear fit equation, and the black values and yellow circles represent the annual mean values of different meteorological factors.

To investigate the relationship between FVC and climatic variables, typical factors such as relative humidity, temperature, sunshine-duration temperature, precipitation, and wind speed were chosen. Maps illustrating the spatial distribution of the correlation between these climate factors and FVC were created. Additionally, correlation image elements significant at the $p < 0.05$ level were extracted, as depicted in Figure 8. In Xinjiang, correlation coefficients linking FVC with rainfall varied between −1 and 1. A marginally stronger positive correlation was evident in 51.24% of the region, and within this subset, 14.52% of the areas that met the 0.05 significance level were primarily situated around the Junggar Basin and to the west of the Tarim Basin. The areas with positive correlation between FVC and rainfall in different ecological functional zones were mainly concentrated in R1, R2 and R3 regions, and the areas with negative correlation, where precipitation is the main limiting factor for vegetation growth, were mainly concentrated in R4 and R5 regions. In Xinjiang, the range of correlation coefficients linking FVC with temperature was between −0.89 and 0.91. A notable negative correlation was observed in 56.06% of the area. Within this, 5.89% of the regions, which successfully met the 0.05 significance level, were predominantly situated in the Hotan, Kashgar, and Kizilsu Kyrgyz Autonomous Prefectures of southern Xinjiang, as well as the Altay and Tacheng regions in the north. Negative correlations were also found in the ecological functional zones of R1, R2, R3, and R4, which is due to the fact that the climate of Xinjiang is dominated by arid and semi-arid climate, and high temperatures may exacerbate water evaporation and thus inhibit growth, whereas in the region of R5 the moisture conditions are good, and the high temperatures promote plant growth, which leads to the positive correlation between the FVC and the temperature.

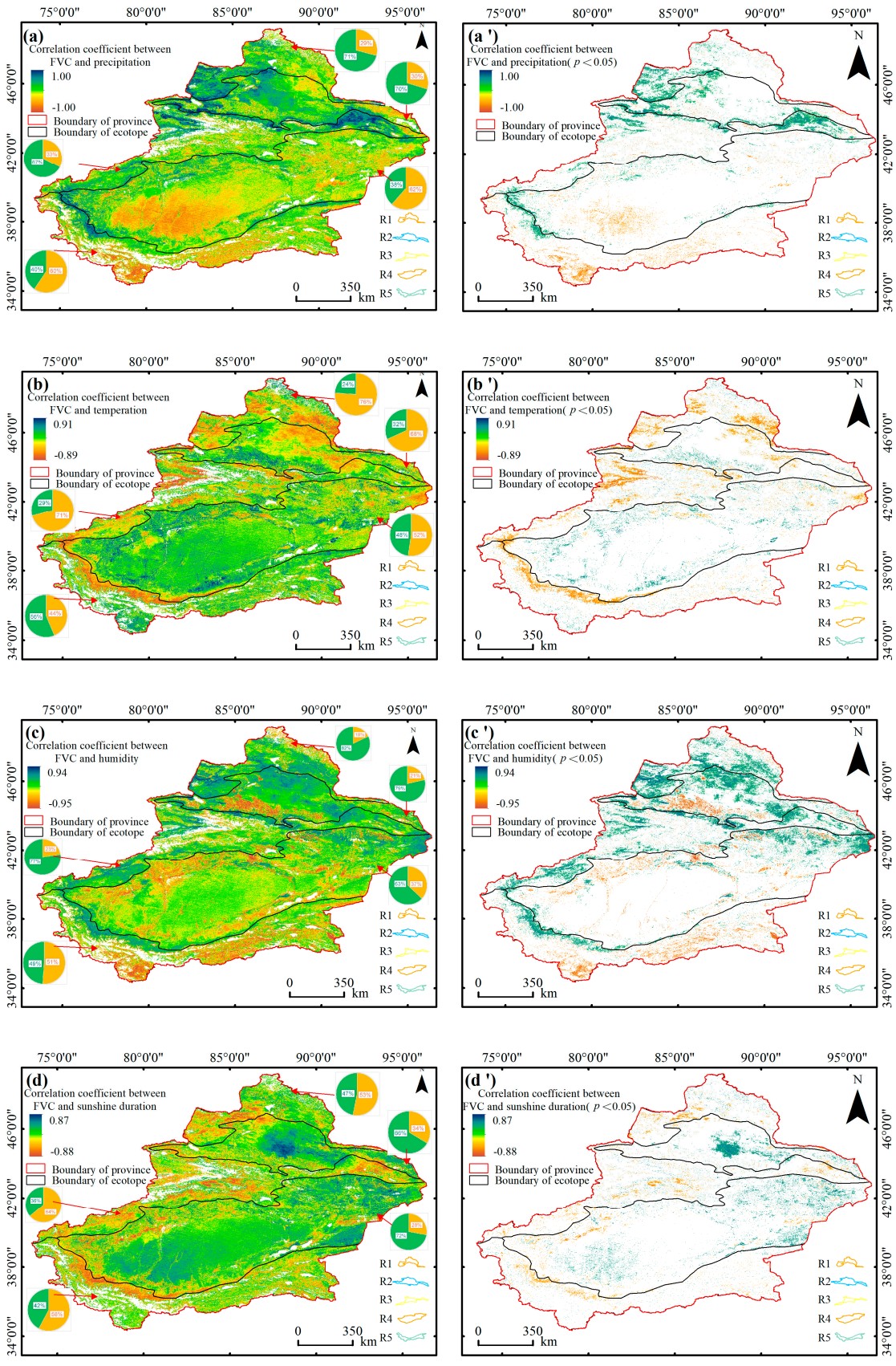

**Figure 8.** *Cont.*

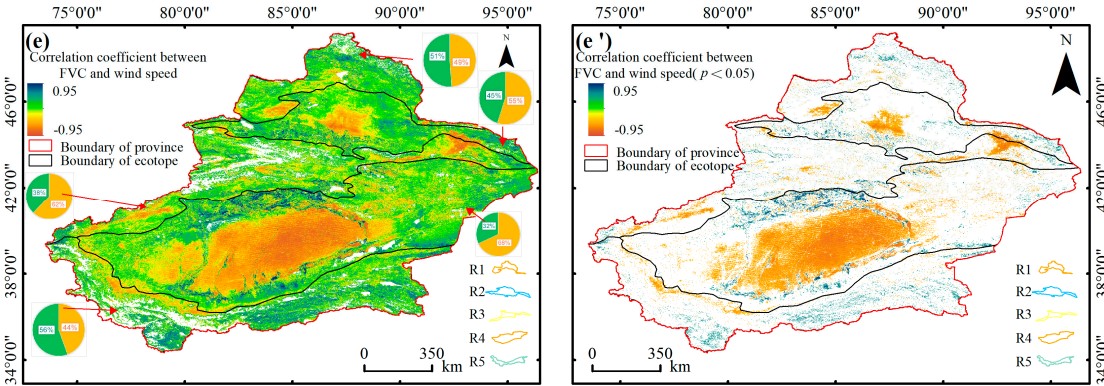

**Figure 8.** Spatial distribution of the correlation between the FVC and climate factors in Xinjiang. (**a–e**) Spatial trend distribution of the FVC vs. precipitation, temperature, relative humidity, daylight hours, and wind speed; (**a′–e′**) significance test of the FVC vs. precipitation, temperature, relative humidity, daylight hours, and wind speed.

In Xinjiang, the correlation coefficients for FVC and sunshine duration varied between −0.95 and 0.94. A notable positive correlation was seen in 58.3% of the region, and out of this, an area constituting 4.22% that met the 0.05 level of statistical significance was predominantly found in the Gurbantunggut Desert; the correlation between the FVC and daylight hours of vegetation in R1, R3, and R5 in different ecological function zones was positive, and sufficient precipitation can generally meet the demand of vegetation growth. Excessive precipitation, on the contrary, would increase cloudiness, which in turn would reduce daylight hours, weakening the photosynthetic efficiency of plants and adversely affecting vegetation growth. From the correlation between the vegetation FVC and relative humidity, the proportion of positively and negatively correlated areas was 67.26% and 32.74%, respectively,15.62% of the areas, primarily in Xinjiang's northern region and the western Kunlun Mountains, exhibited a positive correlation with relative humidity that met the statistical significance threshold of 0.05, indicating that high relative humidity is favorable to the growth of the vegetation FVC in the whole region of Xinjiang and occupies a dominant role. The negatively correlated areas were mainly distributed in R5, while in other ecological functional subregions, the vegetation FVC and relative humidity were positively correlated. The correlation between the mean annual wind speed and vegetation FVC was more negative (59.84%) than positive (40.16%) in the whole region of Xinjiang; a significant negative correlation with wind speed was observed in 16.02% of the areas, predominantly in the Tarim Basin, and this correlation was statistically significant at the 0.05 level, and a positive correlation between wind speed and the vegetation FVC was mainly concentrated in R1 and R5, while a negative correlation between the vegetation FVC and wind speed dominated in sub-zones R2, R3, and R4, where the negative correlation was gradually increasing.

### 3.5. Analysis of Drivers of Vegetation Cover

Exhibited in Figure 9, spatial variability is evident in the enhancement of Xinjiang's vegetation FVC due to climatic variations and human influences. In Xinjiang, regions that have seen an enhancement in vegetation FVC owing to climate change constitute 16.14% of the area. This is primarily observed in the oases located on the Tianshan Mountains' northern and southern flanks, following the order R3 > R2 > R1 > R4 > R5 across various ecological functional zones. Conversely, areas where human interventions predominantly contributed to the FVC increase account for 20.38% of Xinjiang, predominantly in the oases along the northern slopes of the Kunlun Mountains, marked by the sequence R2 > R5 > R3 > R1 > R4 in diverse ecological zones. Additionally, climate change has led to a decline in vegetation FVC in 28.62% of Xinjiang, with the principal impact observed in the Junggar Basin. This decline is arranged as R2 > R1 > R3 > R4 > R5 in the distinct ecological functional

regions. Furthermore, the degradation of vegetation FVC due to human activities covers 15.62% of Xinjiang, displaying a pattern of R1 > R3 > R2 > R5 > R4 in different ecological functional areas.

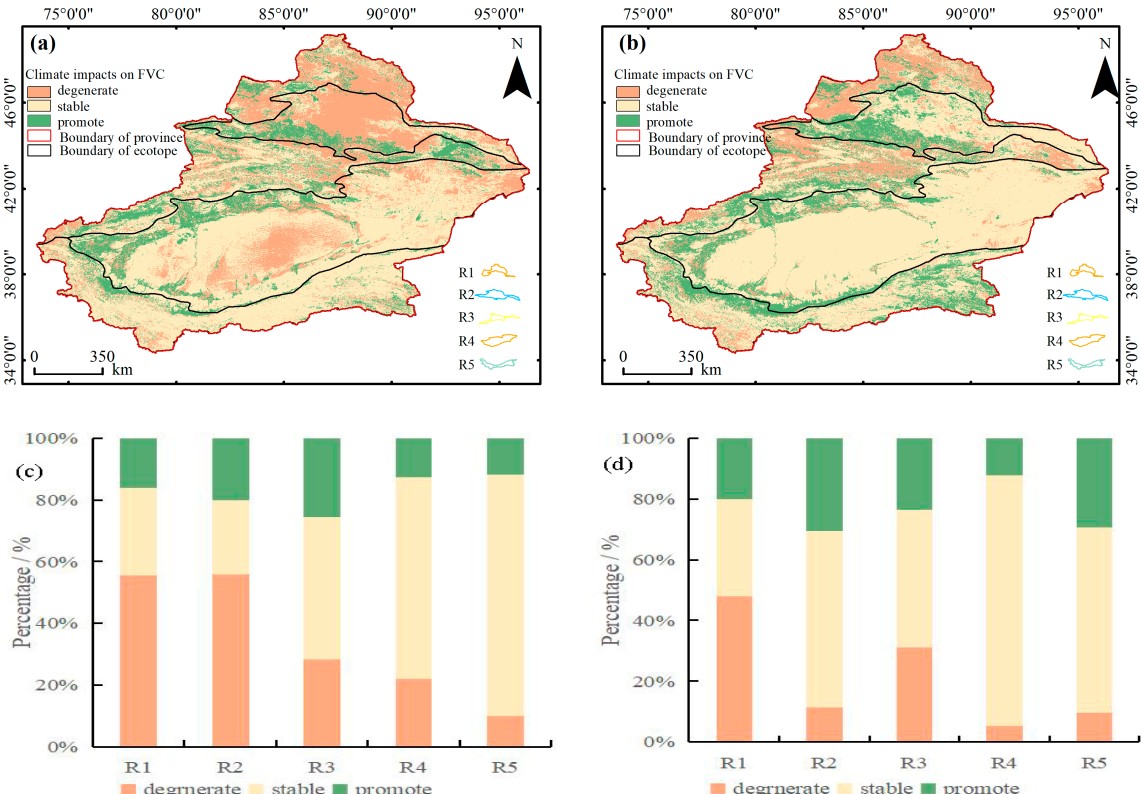

**Figure 9.** (**a**) Spatial distribution of climate change impacts on the FVC; (**b**) spatial distribution of anthropogenic impacts on the FVC; (**c**) statistical map of climate change impacts on the FVC; and (**d**) statistical map of anthropogenic impacts on the FVC.

### 3.6. Relative Contribution of Climate Change and Human Activities to Changes in Vegetation FVCs

In our study, we analyzed how climate change and human activities each contribute to alterations in Xinjiang's vegetation. This was done by evaluating the trends in actual FVC measurements, alongside their predicted values and discrepancies (Table 3). It was found that climate change was responsible for 56.93% of the vegetation changes, while human influences accounted for 43.07%. As illustrated in Figure 10, over 60% of the enhancement in vegetation cover in Xinjiang can be attributed to climate change, affecting 50.72% of the region. This change is primarily noticeable in the Junggar and Tarim Basins, distant from urban centers. The region's shift towards a warmer and more humid climate has markedly benefited vegetation FVC. Conversely, human activities have played a significant role in over 60% of the area, especially in the oases on the northern and southern flanks of the Tianshan Mountains and the northern side of the Kunlun Mountains. These areas have seen an increase in vegetation cover and improvements in soil and climate conditions. This is largely due to extensive irrigation system development and widespread afforestation and landscaping initiatives.

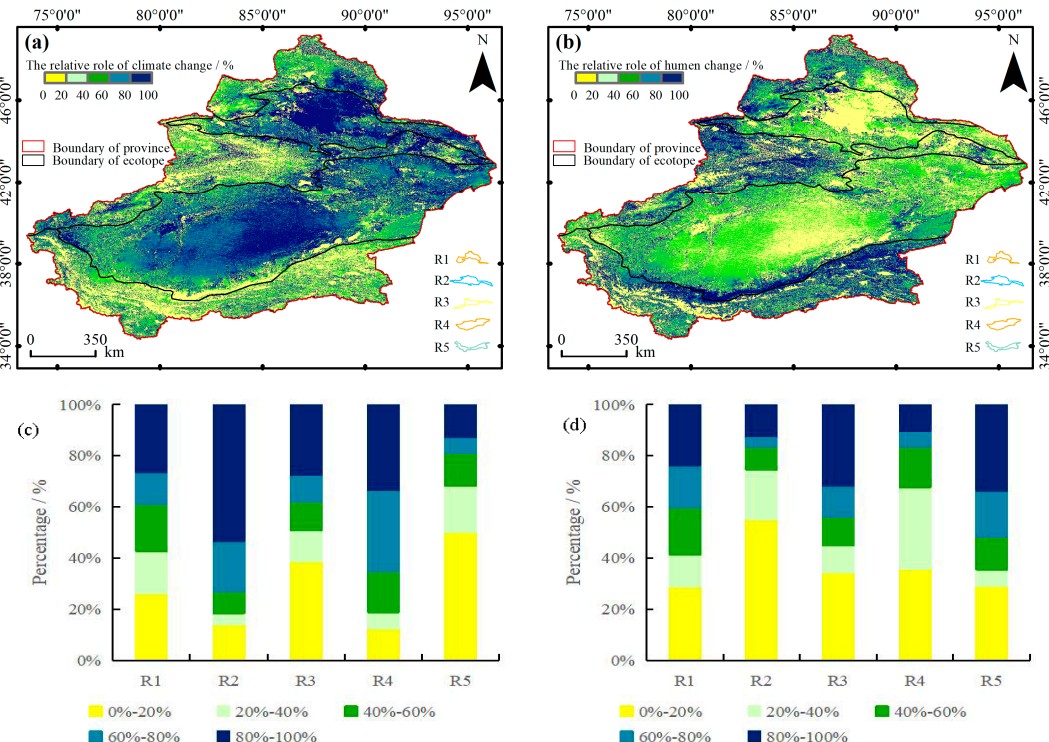

**Figure 10.** (**a**) Spatial distribution of the relative contribution of climate change to the FVC; (**b**) spatial distribution of the relative contribution of human activities to the FVC; (**c**) statistical map of the relative contribution of climate change to the FVC; (**d**) and statistics of the relative contribution of human activities to the FVC.

## 4. Discussion

### 4.1. Characterization of the Spatial and Temporal Evolution of Vegetation FVC

The study's findings revealed a marginal upward trend in Xinjiang's overall FVC from 2000 to 2020, aligning with the outcomes of prior research [54]. The study also noted distinct regional variations in the growth rates of vegetation FVC across Xinjiang, following the sequence R2 > R4 > R5 > R3 > R1. This pattern could be attributed to Xinjiang's unique geomorphology, influenced by climate change, leading to an uneven distribution of heat and precipitation. In Xinjiang, there was a noticeable geographic trend in the vegetation FVC, decreasing gradually from southeast to northwest. This pattern was largely driven by the climatic transition from a warm and moist to a cold and arid environment. However, the oasis agricultural zones in the central Tian Shan and along the Tarim Basin's periphery exhibited a positive growth trend, corroborating the findings of [55]. The growth in the FVC around the Tarim Basin's oasis was linked to increased alpine snow and ice melt. Additionally, the regional government's active involvement in initiatives such as the farmland protection forest project and ecological water conveyance in the Tarim Basin [56] has not only fostered oasis agriculture but also enhanced the oasis's vegetation cover and significantly improved the environmental ecology.

### 4.2. Impacts of Climate Change on Vegetation Dynamics

A recent study has been centered on comprehending the effects of climate change on the FVC patterns within vegetation [57]. Examining the interplay between climatic variables and vegetation FVC is key for tracking vegetation's dynamic shifts and applying effective ecological management strategies. This study delved into the correlations between vegetation FVC trends and various climate factors—including yearly precipitation, average annual temperature, daylight hours, yearly relative humidity, and average annual wind speed—over the period from 2000 to 2020, as illustrated in Figure 8. An overall positive

correlation between the vegetation FVC and precipitation was found; in Xinjiang, the rise in precipitation levels was a contributing factor to the enhancement of the FVC. The positive effect of precipitation in R1, R2, and R3 was significantly higher than that in R4 and R5 and was most pronounced in the Tien Shan Mountain range and the northwestern corner of the northern border [58]. In recent decades, the trend of precipitation in all regions of Xinjiang has been increasing, and the increasing trend of R1, R2, and R3 has been much larger than that of R4 and R5. An increase of precipitation supports the physiological metabolic process of vegetation and promotes its growth and development, which effectively improves vegetation cover. There was a widespread negative relationship between precipitation and the FVC across the entire area, with a notable focus around the borders of regions R4 and R5. This pattern might be attributed to the area's distinct topography, where increased rainfall adversely impacts some plants' metabolic processes and photosynthesis efficiency, subsequently decreasing vegetation cover [59]. In Xinjiang, a generally inverse relationship was observed between air temperature and vegetation FVC; that is, rising air temperatures tended to hinder the growth of FVC in the region [60]. The beneficial effect of air temperature on vegetation FVC was primarily perceived in region R4, particularly in the Tianshan Mountain Range, with the most significant effects seen on the outskirts of R4. Alpine snow and ice and seasonal snowmelt are important sources of water in Xinjiang, and an increase in temperature prompts glaciers and snow to melt, resulting in abundant water resources impacting the vegetation FVC. The inhibitory impact of temperature on the vegetation FVC in Xinjiang was mainly due to the fact that increased temperature exacerbates the evaporation rate of soil moisture in the region, largely counteracting the positive effects of increased temperature on plant photosynthesis [61], thus negatively affecting the vegetation FVC. In general, the relationship between daylight hours and FVC in Xinjiang was found to be positively correlated. This means that a rise in daylight hours positively influenced the growth of vegetation FVC. This effect was particularly pronounced in the R4 region, where increased daylight hours not only promoted the buildup of organic matter in plants but also hastened their growth rate [62]. A negative effect of daylight hours on the FVC in Xinjiang was particularly obvious in southern Xinjiang, mainly because when the daylight hours is too high, plant photosynthesis is weakened or may even stop, impacting the normal growth and development of plants [63]. Relative humidity promoted the growth of the vegetation FVC in Xinjiang and was the dominant factor affecting the vegetation FVC. In regions such as R5, low annual precipitation leads to a decreasing trend of relative humidity in the air, and the reduction of water required for plant growth in turn limits vegetation growth. Optimal wind speeds are beneficial for plant growth and enhancing vegetation cover, whereas excessive or continuous strong winds can harm vegetation and impede its growth. In region R1, the prevailing northwesterly winds carry warm and moist air currents, contributing positively to the development of the vegetation FVC [20].

*4.3. Impact of Human Activities on Vegetation Dynamics*

Concerning the impact of human activities on Xinjiang's FVC between 2000 and 2020, it was observed that these activities predominantly exerted negative effects on the region's vegetation FVC. The negative anthropogenic impacts have mainly been distributed in areas around Tian Shan and Altay Shan in the transition to cropland, which may have been related to excessive anthropogenic deforestation negatively affecting the vegetation FVC [64]. Anthropogenic impacts have also been serious in areas where the oasis edge meets the desert, which are extremely fragile ecological environments where high-intensity anthropogenic reclamation and grazing have caused devastating damage to the vegetation [26]. Human activities have had beneficial effects primarily in oasis agricultural regions and central mountain areas, where advancements in agricultural technology have contributed to the enhancement of vegetation FVC [65]. The central regions of the Tianshan and Altay Mountains, designated as nature reserves, have seen reduced human interference thanks to government-enforced natural forest protection initiatives. This has led to stable

and healthy vegetation growth in these reserves. The increase in vegetation attributed to human influence is most notable in the oases along the northern and southern flanks of the Tianshan Mountains and the northern side of the Kunlun Mountains. Here, significant vegetation cover expansion can be credited to the development of irrigation practices and the execution of ecological management strategies.

*4.4. Research Gaps and Future Prospects*

This study has certain constraints. (1) While the MODIS NDVI is useful for investigating long-term changes in vegetation, its effectiveness is enhanced when integrated with a variety of other satellite datasets and extensive terrestrial long-duration observational data for analyzing variations [66]. (2) The residual error values of the FVC calculated in this study included anthropogenic factors as well as climatic influences [15], and the impact of climate on vegetation change are usually complex. There are also temporary lag effects and cumulative effects to consider [67]. (3) Anthropogenic drivers of vegetation change are both positive and negative [68]. In this research, quantifying the specific impacts of various human activities on vegetation alteration proved to be a complex task, thus necessitating the consideration of human activities in a collective manner [69]. In future studies, field observations and the use of machine learning and other methods should be utilized to determine the contribution of different human activities.

## 5. Conclusions

This research focused on the changing spatial and temporal patterns of the vegetation FVC in Xinjiang from 2000 to 2020, focusing on various ecological functional zones. By employing methods like the Theil–Sen trend analysis, the Mann–Kendall test for significance, and the Hurst index, in combination with correlation assessments, the research explored how vegetation reacts to climatic shifts and anthropogenic factors.

(1) The overall FVC in Xinjiang from 2000 to 2020 showed an upward movement, with a growth rate of $4 \times 10^{-4}$ $y^{-1}$. The FVC in ecological areas showing an increasing trend was indicated by R2 ($1.4 \times 10^{-3}$ $y^{-1}$) > R4 ($8 \times 10^{-4}$ $y^{-1}$) > R5 ($5 \times 10^{-4}$ $y^{-1}$). The FVC in ecological zones showing a decreasing trend was shown as R3 ($-3 \times 10^{-4}$ $y^{-1}$) > R1 ($-1.2 \times 10^{-3}$ $y^{-1}$). In Xinjiang, the FVC's spatial distribution generally showed higher values in the northwest and lower in the southeast, with 66.63% of the area having degraded vegetation and 11% of the area having significant vegetation growth. The future changes in FVC will be dominated by a decreasing trend. In the results of the coefficient of variation, weak variation accounted for 42.12% of the total, the mean variation coefficient recorded was 0.2786, and the stability of different ecological zones was R1 > R3 > R2 > R4 > R5.

(2) Concerning how vegetation and climate aspects affect the area, there was a positive correlation observed between the FVC of vegetation across Xinjiang and factors like relative humidity, precipitation, and sunlight. Conversely, a negative correlation was found with factors such as air temperature and wind speed. Among these, relative humidity had a more pronounced impact on vegetation FVC compared to other weather elements. Additionally, the influence of these climatic variables on vegetation FVC differed across various ecological functional zones in Xinjiang, exhibiting spatial diversity.

(3) In summary, the primary influence on Xinjiang's vegetation FVC alterations is climate change. In the past 21 years, in Xinjiang's growing season, the respective influences of climate change and anthropogenic factors on vegetation FVC shifts were 56.93% and 43.07%. Moving forward, there should be an enhanced focus on monitoring the impact of climate change in the preservation of vegetation.

**Author Contributions:** Conceptualization, G.L. and J.L.; methodology, G.L.; software, G.L. and M.Z.; validation, Y.S.; formal analysis, J.L.; investigation, J.W.; resources, S.W.; data curation, G.L.; writing—original draft preparation, G.L. and J.L.; writing—review and editing, J.F.; visualization, J.L.; supervision, J.F.; project administration, S.W.; funding acquisition, J.F. All authors have read and agreed to the published version of the manuscript.

**Funding:** This work was supported by the Key Research and Development Program of Xinjiang Uygur Autonomous Region, China (2022B03030).

**Data Availability Statement:** Data are contained within the article.

**Acknowledgments:** The authors would like to thank the reviewers for their constructive comments and suggestions.

**Conflicts of Interest:** The authors declare no conflicts of interest.

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
