# Peer review of "Characteristics and Drivers of Vegetation Change in Xinjiang, 2000–2020"

_forests, doi:10.3390/f15020231_

Round 1
Reviewer 1 Report
Comments and Suggestions for Authors
The paper “Characteristics and drivers of vegetation change in Xinjiang, 2000-2020” is based on a very important aspect the following suggestions to be implemented for improving the paper:
(a) Figures are very poorly presented, Kindly improve the figures
(b) Aspect ratio the maps need to be kept same
(c) Refer following papers for better refresentation’
1. Spatio-temporal changes in NDVI and rainfall over Western Rajasthan and Gujarat region of India. Journal of Agrometeorology, 20(3), 189-195.
2. Implications of watershed management programs for sustainable development in rural scenario—a case study from foothills of Punjab state, India. Water Conservation Science and Engineering, 7(4), 647-655.
3. Ecotope-Based Diversity Monitoring of Wetland Using Infused Machine Learning Technique. Water Conservation Science and Engineering, 8(1), 38.Assessment of land cover and land use change impact on soil loss in a tropical catchment by using multitemporal SPOT‐5 satellite images and R evised U niversal Soil L oss E quation model. Land degradation & development, 29(10), 3440-3455.
4. Association between drought and agricultural productivity using remote sensing data: a case study of Gujarat state of India. Journal of Water and Climate Change, 11(S1), 189-202.
5. Dynamic trend of land degradation/restoration along Indira Gandhi Canal command area in Jaisalmer District, Rajasthan, India: a case study. Environmental Earth Sciences, 78, 1-11.
(d) Present the rainfall graph in a better way and present the standard diviation over the years
(e) Use a high resolution DEM to represent the topography of the region
(f) Methodology needs better explanation poorly written
Comments on the Quality of English Language(g) Over all the paper need a lot improvement in writing style and presentation
Reviewer 2 Report
Comments and Suggestions for Authors
Dear Authors,
A well written paper. I have one fairly major issue that I think you need to address and a few minor ones.
Major.
As far as I can see you are associating all the error from your model to human activities. You have no discussion on the errors in the meteorological data or uncertainties in the interpolation process (or for future climate any uncertainty in those predictions). If, as seems reasonable, humans are involved in the process, why do you not include "human population" into you model and see if there is some critical density that affects vegetation change? I would expect very high population density to represent towns and be more or less irrelevant, but low to moderate density might reflect arable (moderate) or livestock (low) farming activities.
Minor
1) most of the "interesting" changes appear at the boundaries between regions. At one point in the paper you discuss "ecotones" or zones of transition, I think that this needs more emphasis, earlier in the paper.
2) there is no indication of the statistical significance of the meteorological factors or how they very across the region.
3) figure 9 a and b have the same legend but different maps
4) line 224 ... "multiple" not "binary" perhaps?
5) legend on figure 1 should have R1, R2 etc
6) Table 1, replace the Chinese character
7) Is Hurst significantly more powerful than a simple auto-correlation?
8) if you have data on "protected areas" and "restoration zones" etc then I would look (and report) on whether changes within those zones were significantly different from the surrounding areas.
Reviewer 3 Report
Comments and Suggestions for Authors
Dear Author, the paper is fairly well-prepared. Please consider the comments below which are in moderate importance.
1. Line 16, please follow the abbreviation rule. When first mention, please write the long form first and abbreviation in brackets as in line 19. Then only use the abbrevation, but follow this rule for both abstract and main text. Please check the other abbreviations.
2. Line 21, before writing the results in the the abstract, it would be better to give brief details about the data sets and methods used in the study.
3. Line before this sentence please write what R1, R3, R2, R5, R4 mean.
4. Line 62, it is not common to use “on” and “in” together. Please revise the sentence.
5. Line 105, it is the first time that you mention NDVI, please follow the abbreviation rule. Besides, in the introduction, a literature should be given about the remote sensing based NDVI analysis related to your application. The literature is very reach about these kind of applications. Please also state your difference compared to them based on NDVI analysis.
6. Line 133-136, the borders of the five regions (from R1 to R5) should be shown on the map. Please illustrate them with appropriate polygons.
7. Line 185-186, please explain how you determined each threshold for each class.If your obtained them from a previous study then cite it. Otherwise, it should be explained how these classes occured.
8. Line 285 and 288, There are 2 Figure 3. Please check and correct.
9. Line 288, please add the explanation of ESD, SD, WSD, WSR, SR, ESR to the caption of the Figure.
10. Line 335, please add the explanation of R-R, R-D, D-D, D-R, R-C, Other to the caption of the Figure.
All the best.
Comments on the Quality of English LanguageThe English quality of the paper is fair enough.
Round 2
Reviewer 1 Report
Comments and Suggestions for Authors
The revision suggested have been implemented by the authors.
Reviewer 3 Report
Comments and Suggestions for Authors
Dear Authors,
Thank you for addressing all my comments. The paper can be published as is.
All the best.